# The association between religious participation and memory among middle-aged and older adults: A systematic review

Bonita Nath[1], Priya Patel[2], Mark Oremus[1] *

**1** School of Public Health Sciences, University of Waterloo, Waterloo, Ontario, Canada, **2** School of Public Health, University of Alberta, Edmonton, Alberta, Canada

* moremus@uwaterloo.ca

## Abstract

### Objectives

Mounting evidence suggests religion plays an important role in maintaining cognition. No prior systematic review has focused on the specific association between religion and the memory domain of cognition in middle-aged and older adults. We carried out a systematic review to explore this association in depth.

### Methods

We searched the PsycINFO, Scopus, and PubMed databases to identify articles assessing any means of measuring religion as the exposure and memory as the outcome. Articles had to report on studies with comparison groups to be eligible for inclusion in the review. We followed the PRISMA checklist to conduct the review (PROSPERO registration # CRD42022330389).

### Results

Nine out of the 1648 citations retrieved in the literature search were included in the review. The majority of included articles had a moderate risk of bias. Most results showed positive associations between religion and memory.

### Discussion

Despite consistency in the direction of association between religion and memory, the literature contained some important research gaps: the studies were cross-sectional; a lack of information existed regarding whether different faiths, sex/gender and depression affected the association; and underpowered studies prevented us from drawing firm conclusions about the direction or magnitude of effect. Longitudinal studies avoiding these issues are needed in this field.

**Data Availability Statement:** All relevant data are within the paper and its Supporting Information files.

**Funding:** The author(s) received no specific funding for this work.

**Competing interests:** The authors have declared that no competing interests exist.

## Introduction

Memory, a key domain of cognition, is known to fluctuate as part of the aging process [1]. Episodic memory, or recollection of specific past events, is particularly susceptible to age-related changes over time [2]. Researchers typically assess these changes using clinical tests such as the Rey Auditory Verbal Leaning Test (RAVLT) [3] or Wechsler Memory Scale (WMS) [4]. While non-pathological, age-related memory loss has been well documented in the literature [5], debilitating memory loss may suggest the presence of major neurocognitive disease such as Alzheimer's dementia [6]. As aging adults comprise the fastest-growing age group globally [7], exploring protective factors for memory loss is of paramount importance to health policy and practice.

As of 2015, 84% of adults worldwide reported identifying with an organized religion [8]. Religiosity or religious involvement, defined as the degree of immersion in religious beliefs or the extent of participation in religious activities [9], has been examined as a potential protective factor against memory decline. According to Hill, religious involvement can promote lifestyle practices that are known to preserve late-life cognition, e.g., abstaining from smoking [10]. Moreover, religious involvement may foster a sense of purpose and meaning to protect against depression [10], which is a known risk factor for cognitive decline [11]. Koenig argues that religious involvement (e.g., Bible study, spiritual meditation) encourages one to think about abstract concepts such as morality and the meaning of life, which stimulate and preserve the higher cortical functions and brain regions involved in memory [12].

A recent systematic review of 17 studies found an overall positive association between religious/spiritual involvement and cognition in older (primarily Christian) adults [13]. However, no prior systematic review has focused specifically on the relationship between religious involvement and the memory domain of cognition. Also, since the Hosseini et al. [13] review was published, newer studies reporting inverse [14] or equivocal [15] associations have appeared in the literature, thereby suggesting the need for an updated review. In fact, a PubMed search of the keywords "religion" and "memory" has shown a steadily increasing number of articles on this topic since 1980, thus suggesting a need to regularly update overviews of this emerging evidence base (S1 Appendix).

The current review aims to summarize, synthesize, and appraise the research on the association between religious involvement and memory in middle-aged and older adults. By exploring the link between a potentially modifiable factor (religious involvement) and memory, this systematic review will provide insights into the means of preserving memory function, a crucial component of healthy aging.

## Methods

### Registration and protocol

The protocol for this systematic review was registered in the PROSPERO International Prospective Register of Systematic Reviews (CRD42022330389). We followed the Preferred Reporting Items for Systematic Reviews and Meta-analyses (PRISMA) guidelines to design, report and conduct the systematic review (S2 Appendix) [16].

### Eligibility criteria

The eligibility criteria for the inclusion of English-language studies in the review were based on the PICOTS framework (population, intervention, comparison, outcome, time, and

setting). 'Intervention' and 'comparison' were changed to 'exposure' and 'non-exposure' for our purposes because the nature of the research question precluded the relevance of randomized controlled trials [17].

Included articles had to contain a comparison group and were therefore restricted to designs such as cross-sectional, case-control, and cohort studies, or derivatives such as nested case-control or case-cohort studies. Included articles had to report the results of primary or secondary, quantitative data analyses involving persons aged 45 years or over. In addition, only studies assessing any measure of religious participation, religiosity or spirituality as the exposure, and the memory domain of cognitive function as the outcome, were included in the review. The comparison group had to be composed of persons who did not have any exposure to religious participation, religiosity, or spirituality.

We excluded articles that did not provide information separately for community-dwelling and institutionalized populations, as well as articles that did not report results separately for persons aged 45 years or over. In addition, review articles, commentaries, case series, case reports, and randomized controlled trials were excluded from the review.

## Information sources

We searched PsycINFO, Scopus and PubMed on November 16, 2022. We started the literature search at the inception of each database. A detailed search strategy is shown in S3 Appendix.

## Selection process

The articles retrieved in the literature search were screened independently by two screeners, using Covidence software [18] at title/abstract and full text screening. Following each screening level, the screeners discussed reasons for including and excluding articles and resolved disagreements by consensus.

## Data extraction

A data extraction form was designed to guide the data extraction process, which involved recording the following information from each article: author(s), year, study design, follow up time, study characteristics (sample size, % female, mean age, country, source population), religion/spirituality (exposure) assessment, memory (outcome) assessment, statistical tests, covariates, results. Study data were extracted by one reviewer and verified for accuracy by a second reviewer.

We reported whatever summary measures of effect were contained in the included articles and classified these articles by memory subdomain category, as follows: working memory, global memory function, verbal episodic memory & short term/immediate and delayed recall memory, subjective assessment of overall memory, and multiple cognitive component tests for memory. Due to the various subdomains of memory described in the articles, we grouped the results by memory type. Table 1 outlines the definitions of the memory subdomains.

## Risk of bias

We assessed risk of bias using a version of the Newcastle-Ottawa Scale (NOS) that was adapted for use with cross-sectional studies [26]. In this version of the NOS, studies were examined in three domains and awarded points based on the extent to which each domain was free of bias. Studies earned a maximum 4 points for the 'selection of sample' domain, maximum 2 points for the 'comparability of the study groups' domain, and maximum 2 points for the 'outcome measurement domain'. In the *comparability* section, studies were awarded 1 point if they

**Table 1. Definitions of memory subdomains used in the studies.**

| Memory subdomains | Definitions |
|---|---|
| Working memory | The domain that assesses cognitive functions such as attention and problem solving that are thus helpful in determining the onset of dementia [19,20]. |
| Global memory function | Subjective cognitive function (SCF) is the subjective assessment of participants' global memory functioning [21]. |
| Verbal episodic memory | The domain of cognitive function involved in storing and retrieving details from past events [22]. |
| Short term/immediate memory | The cognitive domain involved in short term storage of information [23]. |
| Delayed memory | This cognitive domain allows one to gauge their ability to recall information after a period of delay [24]. |
| Subjective assessment of overall memory | The respondents' self-perceptions of their overall memory [25]. |

controlled for age, the most important confounding factor in the association between religion and memory, or one of the following factors instead of age (i.e., sex, gender, education, or marital status), and 2 points for controlling for age and one of these other factors. Studies were classified as low risk (7–8 points), moderate risk (5–6 points), or high risk of bias (4 or fewer points). Two independent reviewers independently assessed the risk of bias for each included article, with disagreements settled by consensus.

## Data synthesis

We used the Synthesis without meta-analysis (SWiM) checklist to narratively synthesize the findings of the included articles [27]. SWiM promotes transparency in systematic reviews that do not contain meta-analyses by encouraging reviewers to report a common set of nine items in their manuscripts. We reported those items from the checklist that were applicable to our review, including the metric of reporting results, study categorization method, results summary, and reporting of data. S4 Appendix contains the completed SWiM checklist. We noted items in the checklist that were not applicable to this review.

## Sources of heterogeneity

We did not conduct a meta-analysis because of the heterogeneity of results in the included articles. Heterogeneity manifested itself in the statistical reporting of results, assessments of exposure, mixes of covariates in regression models, and differences in the eligibility criteria for inclusion of participants. A range of statistics were reported across the studies, e.g., correlation coefficients, *p*-values, regression coefficients, *F*-statistics, and confidence intervals (CIs).

Assessments of exposure included three scores derived from the Duke University Religion Index (DUREL) subscales [28], namely organized religious activity (ORA), non-organizational religious activity (NORA) and intrinsic religiosity (IR); a single score for spiritual wellness obtained from three questions on the Wellness Assessment Tool: found meaning in life, believes spiritual needs are being met, view of spirituality [29]; three scores of religious involvement assessing the frequency of religious attendance across the life-course and currently (more than once a week, once a week, 2 or more times a month, one or more times a year, less than once a year, or not at all), and religious affiliation prior to age 16 years (Conservative Protestant, Mainline Protestant, Other Protestant, Catholic, or other religion) [14]; a single dichotomous variable asking about any participation in religious organizations (yes or no) [30]; a three-level categorical variable asking about the frequency of religious participation in

the last 12 months (daily to weekly, monthly to yearly, or no participation) [15]; a dichotomous variable asking participants whether they prayed privately in places other than a church or synagogue (yes or no) [31]; three scores derived from frequency of religious attendance (not at all to more than once a week), frequency of private prayer (never to more than once a day), and religious belief (higher values equating to stronger religious belief) [32]; a dichotomous variable asking about general participation in religious activities (yes or no) [33]; and frequency of religious service attendance (more than once a week, 2–4 times a month, once a month or less, or never) [34].

A range of covariates were reported in the regression models including age, sex, gender, marital status, wealth, employment status, education, as well as others. S5 Appendix contains all the covariates.

## Summary forest plot

In place of a meta-analysis, we generated a forest plot in R Statistical Software v4.2.2 [35] to summarize the overall strength and direction of effect for each of the nine included articles (Fig S6.1 in S6 Appendix). The sizes of the regression coefficients ($\hat{\beta}$) in the forest plot were proportional to the sample sizes of the included articles.

For Kraal et al. [32], we used the 'aggregate' function in R's metafor package to combine the regression coefficients for the three types of religion into one effect estimate. The 'aggregate' function was also employed as follows: (1) to combine life-course and current religious attendance, and the three memory outcomes, in Hill et al.'s [14] study into one effect estimate; (2) to merge the regression coefficients for immediate and delayed recall, age, and levels of religious attendance into a single summary measure of 'any' versus 'no level' of religious participation in Hosseini et al.'s [15] study; and (3) to combine the regression coefficients for all levels of change in religious attendance from early to mid-life in the fully adjusted model of verbal memory into a single dichotomous measure of 'any' versus 'no change' in Nelson et al.'s [34] article. We assumed a within-study correlation across regression coefficients of 0.5.

To obtain standard errors of the regression coefficients for the studies that only reported p-values (i.e., p < 0.001) and no standard errors or confidence intervals [28,30,36], we conservatively assumed the actual p-value of the findings was 0.001 and utilized the 'se.from.p' function in R's dmetar package to obtain the standard errors.

We took the absolute values of $\hat{\beta} < 0$ when higher scores on memory measurement instruments indicated poorer outcomes [30,36] to standardize the direction of effect in the forest plot (i.e., $\hat{\beta} > 0$ indicated a positive association between religion and memory in the plot).

## Results

The literature search produced 1648 articles: 167 duplicates were removed, 1481 articles were screened at the title/abstract level, and 79 were screened at the full-text level. We excluded 70 articles during the full-text screening stage for the reasons cited in Fig 1 and included 9 articles in the review.

All of the included articles employed a cross-sectional study design, with sample sizes ranging from 164 [33] to 24,669 [15] and spanning 15 countries. When available, the primary measure used in our synthesis of study results was the $\hat{\beta}$; the secondary measure was the Pearson's correlation coefficient.

High-level summary results from the forest plot (Fig S6.1 in S6 Appendix) showed that the point-estimated associations between religion and memory were positive in the nine included articles. However, four of the point estimates were relatively small and the confidence intervals

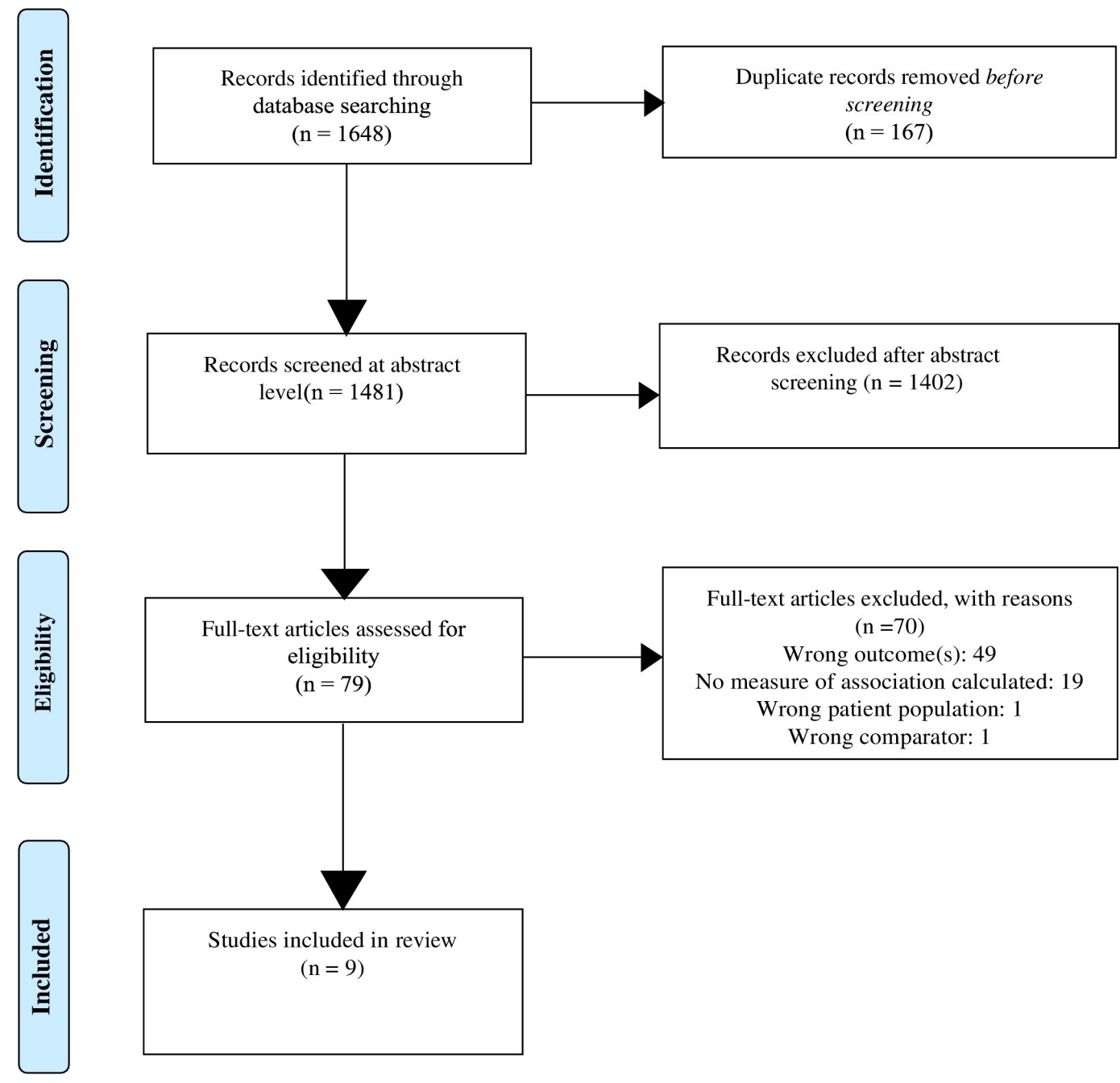

**Fig 1. PRISMA flow diagram of included studies.**

for two of these four estimates included the null value of zero, which indicated uncertainty over whether the true direction of effect in these two articles was positive or negative. The confidence intervals for four other point estimates were fairly wide; this lack of precision adversely affected our ability to judge the magnitude of effect.

Two of the three articles with the largest sample sizes [15,32] had the smallest $\hat{\beta}$s and among the narrowest confidence intervals, although the article with the largest sample size [30] had the second largest $\hat{\beta}$ and the third widest confidence interval (Fig S6.1 in S6

Appendix). Three of the four articles with the smallest sample sizes had the widest confidence intervals [28,31,33]. Three of the five articles with $\hat{\beta}s$ between 0.0 and 0.1 [15,32,36] had sample sizes above 5000, while three of the four articles with the largest $\hat{\beta}s$ (0.38 to 0.50) had sample sizes of 1,135 or less [28,31,33] (Fig S6.2 in S6 Appendix). This suggests an inverse relation between effect size and sample size, as shown by the downward-sloping and dashed line in Fig S6.2 in S6 Appendix. Below we describe specific characteristics and results of the nine included articles, divided into groups defined by the subdomain of memory under investigation.

## Working memory

Only one of the included studies examined working memory. Hill and colleagues [14] investigated the association between religious attendance and memory in later life in 516 community dwelling adults aged 65 years or over in the United States (mean age = 74.10 ± 6.28 years). Religious attendance was divided into life-course religious attendance and current religious attendance. Working memory was tested using the Serial 7s assessment, where respondents are given five trials to subtract 7 from 100 and are scored 0–5 for the number of correct subtractions made [37]. Hill et al. [14] found that life-course religious attendance was negatively associated with working memory $\hat{\beta}$ = -0.067, $p$ = 0.027), whereas the association between current religious attendance and working memory was positive though not statistically significant ($\hat{\beta}$ = 0.057, $p$ = 0.056).

## Global memory function

One study of 164 community dwelling adults aged 65 years or over in South Korea (mean age = 78.21 ± 5.3 years) explored participation in religious activities and global memory function [33]. Participants rated their memory on a 1-question Likert measure of subjective cognitive function (SCF): "What do you think your memory is like now?" [33, p. 1072]. Responses were scored 0–4, with higher scores indicating better memory. The authors found that participation was positively associated with SCF ($\hat{\beta}$ = 0.175, $p$ = 0.013).

## Verbal episodic memory & short term/immediate and delayed recall memory

Six included articles examined verbal episodic memory and short-term/immediate and delayed recall memory. These subdomains of memory were assessed using immediate and delayed word recall tests, where respondents heard a list of words and had to recall as many of the words as possible immediately afterward and again after a 5-minute delay [38].

Four studies specifically explored the association between religion and verbal episodic memory [14,31,32,34]. Lekhak et al. [31] explored the association between private prayer and episodic memory in later life in 1135 community dwelling adults aged 50 years or over in the United States (mean age = 67.5 ± 10.1 years). The authors reported a positive and statistically significant association between private prayer and episodic memory ($\hat{\beta}$ = 0.66, $p$ < 0.01).

Kraal and colleagues [32] assessed religious involvement in 16,089 community dwelling adults aged 51 years or over in the United States (mean age = 68.4 ± 10.2 years). Religious involvement was separated into three different exposure variables: frequency of religious attendance, frequency of private prayer and religious belief. The study found greater improvements in verbal episodic memory among Black and Hispanic respondents engaged in more private prayer ($\hat{\beta}$ = 0.014, p = 0.007) and religious attendance ($\hat{\beta}$ = 0.020, $p$ < 0.001), compared to

White respondents. Additionally, there were higher levels of religious belief linked to lower verbal episodic memory ($\hat{\beta}$ = -0.023, $p$ = 0.004) among Black and Hispanic respondents [32].

In the third study, Hill et al. [14] showed that life-course religious attendance and current religious attendance had non-significant and inverse associations with verbal episodic memory ($\hat{\beta}$ = -0.021, $p$ = 0.050 for life-course; $\hat{\beta}$ = -0.147, $p$ = 0.106 for current). The non-significant associations in the study may be attributed to the small sample of 516 participants.

Nelson et al. [34] examined the frequency of religious attendance in 2,716 community dwelling adults, with an average age of 50 years in the United States. The authors reported a positive association between religious attendance and verbal memory ($\hat{\beta}$ = 0.17; 95% CI = 0.04, 0.30).

Two included studies [15,30] explored links between religious participation and short-term/immediate memory and delayed memory. Hosseini et al. [15] studied a population-level sample of approximately 24,500 community-dwelling adults aged 45 to 85 years in Canada (mean age = 63.0 ± 10.2 years). The authors reported a mix of positive and negative associations following stratification by age and adjustment for multiple sets of covariates, although the regression coefficients were small (-0.09 < $\hat{\beta}$ < 0.06) and most of the 95% confidence intervals included the null value of 0.

Engelhardt et al. [30] investigated the association between participation in a religious organization and 5-minute delayed recall of ten words in 22,949 community-dwelling adults aged 50 to 79 years. The sample spanned 12 primarily European countries (mean age for women = 63 years, mean age for men = 62 years). Participation was associated with better recall ($\hat{\beta}$ = -0.40, $p$ < 0.01) overall, as well as when stratified by males ($\hat{\beta}$ = -0.10, $p$ < 0.01) and females ($\hat{\beta}$ = -0.49, $p$ < 0.01). In this study, negative regression coefficients suggested better cognitive functioning because they were closer to the optimal curve between an individual's age, educational level and their cognitive function [30].

## Subjective assessment of overall memory

Hill et al. [14] investigated the subjective assessment of overall memory by asking participants to rate their memory on a 1–5 scale ranging from poor to excellent, with higher scores denoting better memory. Life-course religious attendance was positively associated with self-rated memory ($\hat{\beta}$ = 0.048, $p$ = 0.015), whereas current religious attendance was negatively associated with self-rated memory ($\hat{\beta}$ = -0.061, $p$ = 0.034) [14].

## Studies with single score outcome derived from multiple cognitive component tests

Some included studies administered batteries of tests to assess multiple subdomains of memory, with the scores aggregated across the individual tests to obtain single summary scores. Jung and colleagues [28] explored the association between three religious variables: ORA, NORA, and IR and memory in 325 outpatients of a psychiatric clinic in South Korea (mean age = 79.15 ± 6.47 years). The authors derived a summary memory score from tests of verbal memory and delayed memory. ORA ($r$ = 0.14, $p$ = 0.010), NORA ($r$ = 0.12, $p$ = 0.040) and IR ($r$ = 0.14, $p$ = 0.012) were positively correlated with memory [28].

Strout et al. [36] investigated the association between spiritual wellness and memory impairment in 5,604 community-dwelling adults aged 60 years or over in the United States (mean age = 83 ± 6.2 years). The authors derived a single score for memory from the Cognitive Performance Scale, which assessed short-term, overall, and procedural memory (higher scores

indicated poorer memory). Mean spiritual wellness was negatively associated with impaired memory ($\hat{\beta} = -0.055$, $p < 0.001$).

## Risk of bias

Overall, one study had a low risk of bias [33] and the remaining eight had a moderate risk of bias (Fig 2 and S7 Appendix). Most of the biases occurred in the selection section of the NOS, given that none of the articles compared characteristics of their respondents and non-respondents and most studies did not perform sample size calculations.

Each study can earn a maximum of 8 stars (*selection*: 4 stars, *comparability*: 2 stars, *outcome*: 2 stars).

## Consideration of the impact of other variables on the association

We also assessed whether any other variables such as age, gender and socioeconomic status had an impact on the association between religious participation and memory in our review. Out of all the included studies, only Lekhak et al. [31] addressed the impact of other variables, namely age and gender, on this association. In the study, it was found that after adjusting for gender, more women reported using prayer and had a higher episodic memory score compared to men ($\hat{\beta} = 1.11$, p < .001) [31]. The authors also noted a slightly higher episodic memory score among those who pray as they become older ($\hat{\beta} = 0.04$, p < .05) [31].

# Discussion

## Summary of evidence

Overall, most studies reported positive associations between religion and memory [14,30,28,33,31,36]. However, we found several gaps in the literature. Since all the studies were cross-sectional, we could not tell whether increased religiosity preserved memory function or if persons with memory impairment withdrew from religious participation. Furthermore, none of the studies assessed mediating factors such as loneliness and, therefore, provided limited information on the underlying components of the mechanism of association between religion and memory. None of the studies assessed the impact of different faiths (or denominations within faiths) on the association between religion and memory. Therefore, we could not assess effect modification by faith. Given the Euro-American settings of most included articles, the samples were likely to be predominantly Christian. This could also be the case for the articles from South Korea because the latter country has a large Christian population. Until more information is known about the association between religion and memory across different faiths, caution must be taken when applying the results of the included studies to non-Christian populations. Moreover, due to the multiple means of measuring exposures and outcomes in the included articles, we could not ascertain which components of religion were optimal protective exposures for memory, nor could we assess which subdomains of memory benefitted the most from religion.

Finally, most of our studies had a moderate risk of bias rating, which entails important implications for the conclusions of the review. Most concerning was that none of the included articles compared the characteristics of respondents to non-respondents, thereby preventing us from assessing the full scope of selection bias and ascertaining whether the persons recruited into the studies were adequately representative of the populations from which they were drawn.

The summary effects in the forest plot showed a positive association between religion and memory overall, though small effect sizes in some instances and wide confidence intervals in

A.

| | Engelhardt et al., 2010 | Hill et al., 2020 | Hosseini et al., 2021 | Jung et al., 2019 | Kim et al., 2021 | Kraal et al., 2019 | Lekhak et al., 2020 | Strout et al., 2015 | Nelson et al., 2022 |
|---|---|---|---|---|---|---|---|---|---|
| **Selection** | ⬤ | ⬤ | ⬤ | ⬤ | ◯ | ⬤ | ⬤ | ⬤ | ⬤ |
| **Comparability** | ⬤ | ⬤ | ⬤ | ⬤ | ⬤ | ⬤ | ⬤ | ⬤ | ⬤ |
| **Outcome** | ⬤ | ⬤ | ⬤ | ⬤ | ⬤ | ⬤ | ⬤ | ⬤ | ⬤ |
| **Total # of stars** | 6 | 6 | 6 | 6 | 7 | 6 | 6 | 6 | 6 |
| **Risk of bias** | moderate | moderate | moderate | moderate | low | moderate | moderate | moderate | moderate |

B.

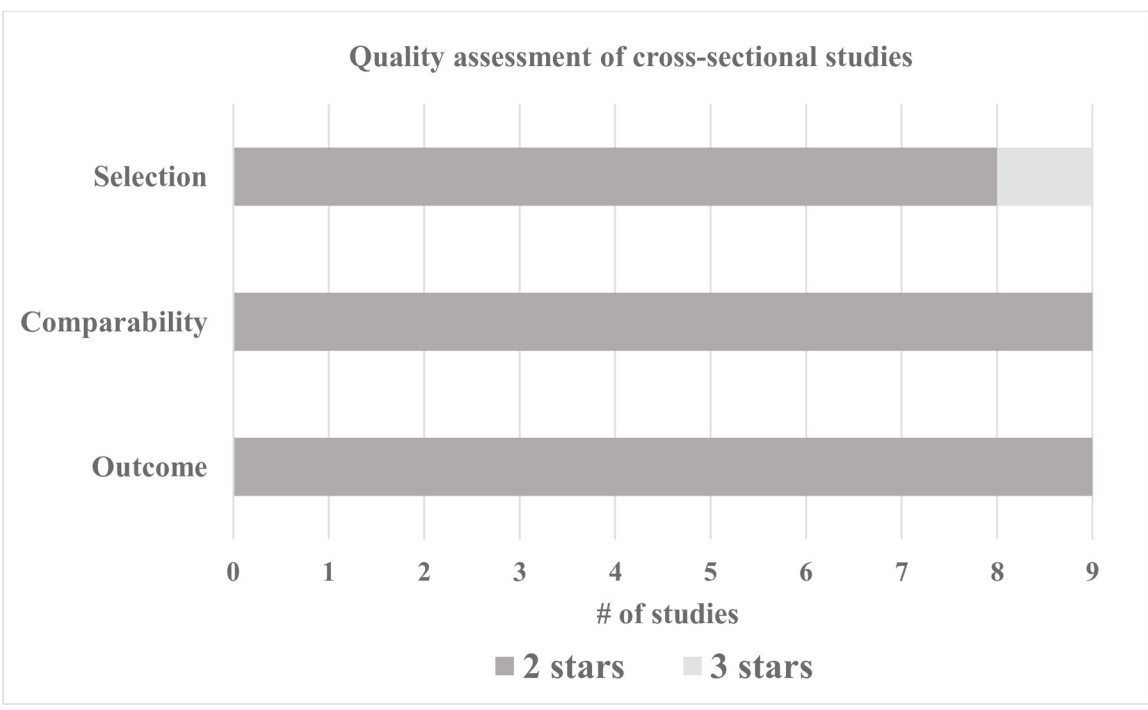

**Fig 2. Adapted Newcastle–Ottawa scale for quality assessment of included articles.**

other instances limited our ability to draw conclusions about the magnitude of the true effect. Out of the 29 specific results reported in the included articles, 11 were not statistically significant. Assuming minimal biases and underpowered analyses, the absence of statistical significance prevented us from firmly concluding whether the overall direction of the true effect was positive.

## Strengths and limitations of the review

We used the 16-item AMSTAR-2 (A MeaSurement Tool to Assess systematic Reviews) to critically appraise the systematic review [39]. We responded affirmatively to the 13 questions that applied to our review (three questions about meta-analysis did not apply), which indicated the review was likely to give a rigorous and thorough summary of results from the included articles (S8 Appendix) [40].

We were unable to conduct a meta-analysis due to the heterogeneity of exposure and outcome measures, which has been observed in previous reviews of religion and cognition [15] and religion and health [41]. To assist in presenting a comprehensive snapshot of the results, we developed a forest plot that relied on certain assumptions about correlations across regression coefficients, p-values, and directions of effect (described under 'summary forest plot' above). The accuracy of the plot depends on the validity of these assumptions. Since we limited the literature search to English-language articles, we may have excluded non-English studies on the topic.

## Future recommendations and conclusion

As a result of gaps in the literature, further exploration is needed to better understand the association between religion and memory. One key item to improve our understanding of the topic would be to conduct longitudinal studies and minimize reverse causality bias and allow us to explore the temporal association between religion and memory. The availability of data from ongoing cohort studies such as the Canadian Longitudinal Study on Aging [42] permits researchers to examine the association longitudinally. Another key item is to explore the association between religion and memory in multiple faiths to observe any significant changes or impact on the strength and direction of effect.

One may also investigate possible sex and gender differences in the association between religion and memory. Some earlier findings suggest that religious engagement may enhance cognitive functioning among older women. For instance, Corsentino et al. [43] found that religious engagement may be protective against global cognitive decline among older women with depressive symptoms. Additionally, Lee et al. [44] showed how religious participation helped to improve cognitive functioning among older women, whereas social activities were found to be more cognitively stimulating among older men. While these studies did not specifically explore the association of religious engagement and memory, their findings provided us with insight on the important role of biological sex on cognitive well-being in later life. Future studies could investigate how the association between religion and memory may vary for women and men, and whether engagement in specific religious activities may entail greater benefits for females versus males, or vice versa.

Other items to explore would be the potential mediating or moderating roles of factors such as depression, social support, or loneliness in the relation between religion and memory. Depression has strong associations with both religion and memory, particularly among older adults, though its moderating or mediating role has not been thoroughly explored in this area. Religious involvement has been linked to hope, resiliency, and the ability to cope with adversities among depressed persons [45], and depression has been found to increase the risk for impaired memory [46]. Functional social support has been shown to be positively associated with memory [47], and social isolation/loneliness have been shown to be inversely associated with memory [48]. Given the links between memory function and major neurocognitive disorder, these forms of social connectedness may also be associated with dementia [47,49]. Religious involvement may promote functional social support and decrease loneliness/social isolation [50,51].

Our systematic review provided a comprehensive overview of research on the roles of religion and memory in later life. Most included studies reported positive associations between religion and memory [14,30,28,33,31,36], as shown by the summary effects in Fig S6.1 in S6 Appendix. However, we could not draw definitive conclusions about the true direction or magnitude of effect. This was because of wide confidence intervals in some instances, moderate levels of risk of bias in eight of nine studies, and cross-sectional designs. Further, the included studies contained some important research gaps by not addressing whether different faiths, sex/gender and depression affected the association of interest. Additionally, many studies did not adequately handle prognostic factors, as only four studies excluded persons with neurological conditions [15,28,30,33] and five studies adjusted for depressive symptoms in regression analysis [14,15,32,33,34] (S5 Appendix). While stronger evidence from further studies is needed to fill current research gaps, the review highlighted the state of the published literature on this topic.

## Supporting information

**S1 Appendix. Number of articles on religion and memory, published annually from year 1980 to 2022.**
(DOCX)

**S2 Appendix. Preferred reporting items for systematic reviews and meta-analyses 2020 Checklist.**
(DOCX)

**S3 Appendix. Search phrases.**
(DOCX)

**S4 Appendix. Synthesis without meta-analysis (SWiM) reporting items.**
(DOCX)

**S5 Appendix. Data extraction of included articles.**
(DOCX)

**S6 Appendix. Summary plots of the associations between religion and memory.**
(TIF)

**S7 Appendix. Adapted Newcastle–Ottawa scale for quality assessment of cross-sectional studies.**
(DOCX)

**S8 Appendix. AMSTAR 2: A critical appraisal tool for systematic reviews that include randomised or nonrandomised studies of healthcare interventions, or both.**
(DOCX)

## Acknowledgments

We would like to express our thanks to Jackie Stapleton for her assistance with developing the search strategy for this review. The authors also wish to thank Karl Grewal for helping us to classify the included studies by memory domain category.

## Author Contributions

**Conceptualization:** Bonita Nath, Mark Oremus.

**Data curation:** Bonita Nath, Priya Patel.

**Formal analysis:** Bonita Nath, Priya Patel.

**Methodology:** Bonita Nath, Mark Oremus.

**Project administration:** Mark Oremus.

**Resources:** Mark Oremus.

**Supervision:** Mark Oremus.

**Writing – original draft:** Bonita Nath, Priya Patel.

**Writing – review & editing:** Priya Patel, Mark Oremus.

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
