## [Decision Letter · Decision Letter 0]

5 Jun 2023

PONE-D-23-08924The association between religious participation and memory among middle-aged and older adults: a systematic reviewPLOS ONE

Dear Dr. Oremus,

Thank you for submitting your manuscript to PLOS ONE. After careful consideration, we feel that it has merit but does not fully meet PLOS ONE’s publication criteria as it currently stands. Therefore, we invite you to submit a revised version of the manuscript that addresses the points raised during the review process.

ACADEMIC EDITOR:Thank you for submitting your work to PLOS ONE. We have finished an initial round of review and have received insightful and important critiques from our external experts. Please see them below and address them. While we see value in the work conducted and commend the authors for their diligence and methodological rigor, there are major limitations. We would like to see your revised work and your responses to these limitations and questions before making a final decision.

We look forward to receiving your revised manuscript.

Kind regards,

Fares Alahdab, MD, MSc

Academic Editor

PLOS ONE

Reviewers' comments:

Reviewer's Responses to Questions

**Comments to the Author**

1. Is the manuscript technically sound, and do the data support the conclusions?

Reviewer #1: Yes

Reviewer #2: Yes

Reviewer #3: Partly

2. Has the statistical analysis been performed appropriately and rigorously? 

Reviewer #1: Yes

Reviewer #2: Yes

Reviewer #3: No

3. Have the authors made all data underlying the findings in their manuscript fully available?

Reviewer #1: Yes

Reviewer #2: Yes

Reviewer #3: No

4. Is the manuscript presented in an intelligible fashion and written in standard English?

Reviewer #1: Yes

Reviewer #2: Yes

Reviewer #3: Yes

5. Review Comments to the Author

Reviewer #1: The present article provides a systematic review of studies that examine the relationship between religious activity and memory function in middle-aged and older adults. The authors searched for articles and included nine studies that met the eligibility criteria, which included a comparison group enabling valid statistical tests. Due to the heterogeneity of results in the included studies, the authors did not perform a meta-analysis but instead evaluated the quality of the studies and reviewed them individually. They concluded that while most studies showed a positive association between religious activity and memory function, there are several deficits in the current studies. Based on their review, the authors suggest that “longitudinal studies with adequate sample sizes and assessment of relevant mediators and moderators are needed in this field” to quote their words.

The article's strong point is its methodology. The authors demonstrate a high level of expertise in conducting a meta-study, with clear and adequate methodological details.

However, one major issue with this article is that the take-home message throughout the main text and especially for figures is hard to grasp. While the authors included the quality check of studies as a main figure, they only attached the summary of statistics, which is a forest plot of beta coefficients, as a supplementary figure. Although the decision not to conduct a meta-analysis due to the heterogeneity of results in the included articles is understandable, a main figure that effectively conveys the characteristics of the reviewed studies, such as consistency in positive association, small effect size, and wide confidence interval, is necessary. I suggest that the authors consider presenting an adequate figure, such as a comparison of effect size and sample size, to effectively convey the characteristics of the reviewed studies.

Specific comments:

As the authors noted, the confidence intervals for four point estimates were wide, making it difficult to determine the magnitude of the effect. As a systematic review, the source of such large variability should be noted. One potential source is the sample size, as most studies did not perform sample size calculations. The authors could compare the sample sizes of the studies in the main text, although the exact numbers are specified in the supplementary materials. This would provide valuable information and help readers better understand the results and limitations of the reviewed studies.

Reviewer #2: Summary:

In this manuscript the authors completed a systematic review of the literature to answer the question of the association between religious participation and memory among middle-aged and older adults. After applying exclusion criteria, only 9 articles were included. All the included articles employed a cross-sectional study design. The majority of included articles had a moderate risk of bias. Most results showed positive associations between religion and memory.

Comments:

- The hypothesis tested in this manuscript consists of heterogeneous and confounded exposures and outcomes. Religious practice/ spirituality includes different religions/ and practices which made the exposure studied heterogeneous. Also, the outcome (different types of memory measures) is also heterogeneous and can be confounded by normal aging, neurodegeneration, sleep disorders, psychiatric disorders, neurologic disorders etc. It is not clear if the included studies accounted for these clinical diagnoses in their study design and thus the conclusion from the systematic review may not be accurate.

- The authors did a good job showing a high degree of bias in the included studies. They also showed that the effect size and association as shown in the forest plot have non-significant and imprecise associations/ estimates. This led rightfully to the inference that current evidence is not sufficient to make any conclusion about the analyzed hypothesis.

- One main confounder is loneliness. Loneliness has been strongly associated with increased risk for dementia (which affects memory typically). Religious practices usually increase sense of community and decrease loneliness, and this could be what drives the association in positive studies. Please review: https://n.neurology.org/content/98/13/e1337

- The authors did a good job discussing how the association is based on cross sectional studies and how reversed causality might be confounding the association.

- The methods section and analyses seem well designed and comprehensive.

Conclusion:

- As a systematic review, this manuscript seems to do a good job in summarizing the literature reaching a conclusion that more research is needed and current evidence is weak and biased.

- My issue is not with the methods/ results, my issue is with the hypothesis itself. A very heterogeneous “exposure” is been studied for its association with a very heterogeneous outcome “different memory measures” in participants with unknown clinical data and in cross sectional studies that are biased and weak. Again, loneliness could be the main mediating factor here (if the association stands). Thus, the results and conclusion were predictably non-decisive and just showed the need for more and better research. Thus, I am not sure how much this manuscript adds to the literature, except that it provides an updated summary of weak studies.

Reviewer #3: This paper presents a systematic review exploring the association between religion and memory. Recognizing the role of religious involvement in cognitive health, the study analyzes a selection of cross-sectional studies to glean insights into the potential protective qualities of religion on memory. The researchers found a generally positive correlation between religion and memory, but also identified key limitations and gaps in the current body of literature, such as lack of diversity in faiths considered and inability to establish causality due to the cross-sectional nature of the studies. The findings underscore the need for more rigorous and varied research to further understand this complex relationship, underlining the paper's contribution to identifying areas for future exploration within this topic.

The methodology of the paper entails a systematic review of existing literature, using the Newcastle-Ottawa assessment tool to critically appraise the included articles. This approach has both strengths and weaknesses.

-Strengths

1 Systematic Approach: The use of a systematic review approach is commendable as it enables a comprehensive exploration of the existing literature, which provides a reliable and robust overview of the field of study.

2 Use of AMSTAR-2: Using the AMSTAR-2 tool for critical appraisal indicates an attention to rigor and thoroughness in the review process. This tool is recognized for its capacity to assess the quality of systematic reviews, hence enhancing the validity of their findings.

3 Identification of Gaps: The methodology was successful in identifying key gaps in the existing literature, such as a lack of longitudinal studies and an underrepresentation of faiths other than Christianity. Recognizing these gaps is crucial for directing future research and deepening our understanding of the subject matter.

-Weaknesses

1 Selection Bias: The authors note a lack of comparison between respondents and non-respondents in the included studies, which prevents a full understanding of selection bias. This lack of comparison could skew the conclusions of the review, as it could fail to account for any systematic differences between these two groups.

2 Lack of Longitudinal Studies: All studies reviewed were cross-sectional, which means the researchers could not determine cause-and-effect relationships, only correlations. Longitudinal studies would have allowed for a more nuanced understanding of how religious involvement affects memory over time.

3 Lack of Diversity in Faiths: The sample of reviewed studies was predominantly set in Euro-American contexts, likely skewing towards Christian populations. This limits the generalizability of the findings to other religious groups.

4 Heterogeneity of Exposure and Outcome Measures: The various ways of measuring exposures (religion) and outcomes (memory) in the included studies made it difficult to ascertain specific correlations. This high level of heterogeneity also prevented the authors from conducting a meta-analysis.

Suggestions for Improvement

1 Expanded Inclusion Criteria: The authors could consider including non-English language studies to diversify the cultural and religious contexts of their data. This could potentially yield more nuanced and globally applicable insights.

2 Conduct a Meta-Analysis: If possible, it would be beneficial to standardize or categorize the different exposure and outcome measures to allow for a meta-analysis. This could provide more quantitative insights into the magnitude and direction of the effect of religion on memory.

3 Collaboration for New Research: Given the identified gaps in longitudinal data and diversity of faiths, the authors could collaborate with other researchers or institutions to conduct new, well-designed longitudinal studies. These should aim to fill the highlighted gaps, including various religious denominations and more diverse geographical and cultural contexts.

4 Consideration of Other Variables: The authors might consider the potential impact of other variables such as age, socioeconomic status, and education. Accounting for these variables could help to explain some of the observed associations between religion and memory.

5 In-depth Analysis of Non-significant Results: It could be helpful to discuss the non-significant results in more depth. This could potentially offer important insights into the circumstances or factors that might mitigate the association between religion and memory.

-Assessment of the statistical analysis methods in this SR:

The authors of this paper have encountered a common issue when conducting systematic reviews - the heterogeneity of exposure and outcome measures, and differences in statistical reporting across the included studies. This makes a meta-analysis difficult to carry out as pooling together such diverse data might lead to misleading or inaccurate conclusions. To manage this, they have generated a forest plot, which provides a visual summary of the strength and direction of the effects of the included studies.

In order to create a meaningful forest plot, the authors have combined various exposure and outcome measures to generate single effect estimates. For instance, they used the 'aggregate' function in R’s metafor package to combine the regression coefficients for three types of religion into one effect estimate in the study by Kraal et al. This approach is valid but, as the authors have noted, it introduces an element of assumption, specifically the assumption of a within-study correlation across regression coefficients of 0.5. In the absence of actual data, this assumption might be reasonable, but it also introduces a degree of uncertainty into their results.

Additionally, the authors have made some assumptions to deal with missing data, which is another common problem in systematic reviews. Specifically, they've estimated standard errors for studies that only reported p-values but not standard errors or confidence intervals. They've done this by conservatively assuming the actual p-value of the findings was 0.001 and using the ‘se.from.p’ function in R’s dmetar package. This approach seems prudent, but it does rely on an assumption that may or may not be accurate.

Lastly, the authors standardized the direction of effect in the forest plot by taking the absolute values of negative regression coefficients when higher scores on memory measurement instruments indicated poorer outcomes. This ensures that a positive regression coefficient consistently indicates a positive association between religion and memory in the plot. It's a reasonable approach to standardize the interpretation of the results, but it's worth noting that it involves manipulating the original data, which could potentially obscure some of the nuances of the original findings.

In summary, while the authors' approach to dealing with the heterogeneity of the studies and missing data is innovative and careful, it does introduce a degree of uncertainty into their results. The use of assumptions in their analysis means that the true effect sizes might be somewhat different from what they've reported. Future research in this area would benefit from more consistent reporting of statistical results and measures in primary studies, which would enable a more precise and confident synthesis of the evidence.

6. PLOS authors have the option to publish the peer review history of their article (what does this mean?). If published, this will include your full peer review and any attached files.

Reviewer #1: No

Reviewer #2: No

Reviewer #3: No

---

## [Author Response · Author response to Decision Letter 0]

10 Jul 2023

Please refer to the uploaded 'Response to Reviewers' document to read our responses to each reviewer's comments.

---

## [Decision Letter · Decision Letter 1]

26 Jul 2023

PONE-D-23-08924R1The association between religious participation and memory among middle-aged and older adults: a systematic reviewPLOS ONE

Dear Dr. Oremus,

Thank you for submitting your manuscript to PLOS ONE. After careful consideration, we feel that it has merit but does not fully meet PLOS ONE’s publication criteria as it currently stands. Therefore, we invite you to submit a revised version of the manuscript that addresses the points raised during the review process.

ACADEMIC EDITOR:Thank you for revising your manuscript, it has improved. However, the reviewers still point to a few aspects that require more attention. Please see their concerns below. We look forward to considering your revised work.

We look forward to receiving your revised manuscript.

Kind regards,

Fares Alahdab, MD, MSc

Academic Editor

PLOS ONE

Reviewers' comments:

Reviewer's Responses to Questions

**Comments to the Author**

1. If the authors have adequately addressed your comments raised in a previous round of review and you feel that this manuscript is now acceptable for publication, you may indicate that here to bypass the “Comments to the Author” section, enter your conflict of interest statement in the “Confidential to Editor” section, and submit your "Accept" recommendation.

Reviewer #1: (No Response)

Reviewer #2: (No Response)

2. Is the manuscript technically sound, and do the data support the conclusions?

Reviewer #1: Yes

Reviewer #2: Partly

3. Has the statistical analysis been performed appropriately and rigorously? 

Reviewer #1: Yes

Reviewer #2: Yes

4. Have the authors made all data underlying the findings in their manuscript fully available?

Reviewer #1: Yes

Reviewer #2: Yes

5. Is the manuscript presented in an intelligible fashion and written in standard English?

Reviewer #1: Yes

Reviewer #2: Yes

6. Review Comments to the Author

Reviewer #1: One major concern I raised in my initial review was that the “take-home message throughout the main text and especially for figures is hard to grasp”. Unfortunately, I did not find any specific response addressing this issue in the "response to reviewers." I kindly request the authors to reconsider the clarity of the take-home message and ensure that it is effectively conveyed in the main figures. While the authors have revised the supplementary figure, I still strongly recommend incorporating the relevant contents of Fig. S.6 into the main figure to better present their findings. One possible approach could be to include Fig. S.6 into the Figure 3.

I appreciate the authors' efforts in responding to the “specific comments”, especially in terms of clarifying the description of confidence intervals, the scale of β estimates, and the sample size. These revisions indeed support the conclusion “The summary effects in the forest plot showed a positive association between religion and memory overall, though small effect sizes in some instances and wide confidence intervals in other instances limited our ability to draw conclusions about the magnitude of the true effect.”.

Minor Points:

- In Figure S6.2, I noticed that the x-axis label indicates "(95% confidence interval)”, despite there is no confidence interval displayed.

Reviewer #2: The manuscript was partially improved with the new edits.

However, still one more edit I think is very important.

The authors in their reply mentioned:"

Engelhart et al. excluded those who reported a

stroke, cerebral vascular disease, Parkinson's

disease, or those who were taking medications

for anxiety or depression, or persons who had

received treatment in a psychiatric unit; Hill

et al. controlled for depressive symptoms;

Hosseini et al. excluded individuals with

cognitive impairment and controlled for

depressive symptoms. Given that we only

acknowledged when authors accounted for

these items as covariates, we have added

information on exclusion criteria to the S5

Appendix."

I think this should be clearly mentioned in the discussion section, and they should add that most studies did not account for neurocognitive disorders, dementias or neurodegenerative disorders that clearly affect memory (which include Alzheimer, hippocampal sclerosis, vascular related cognitive decline, FTD, LBD, PD etc). This missing information makes the results of any of the included studies uninterpretable. The authors should clearly say that and clearly say that the conclusions of the meta-analysis is very weak and cannot be accurately trusted knowing that the included studies all have very serious flaws.

7. PLOS authors have the option to publish the peer review history of their article (what does this mean?). If published, this will include your full peer review and any attached files.

Reviewer #1: No

Reviewer #2: No

---

## [Author Response · Author response to Decision Letter 1]

2 Aug 2023

Please see our cover letter and the attached 'response to reviewers'.

---

## [Editor Report · Decision Letter 2]

7 Aug 2023

The association between religious participation and memory among middle-aged and older adults: a systematic review

PONE-D-23-08924R2

Dear Dr. Oremus,

We’re pleased to inform you that your manuscript has been judged scientifically suitable for publication and will be formally accepted for publication once it meets all outstanding technical requirements.

Kind regards,

Fares Alahdab, MD, MSc

Academic Editor

PLOS ONE

---

## [Editor Report · Acceptance letter]

9 Aug 2023

PONE-D-23-08924R2 

The association between religious participation and memory among middle-aged and older adults: a systematic review 

Dear Dr. Oremus:

I'm pleased to inform you that your manuscript has been deemed suitable for publication in PLOS ONE. Congratulations! Your manuscript is now with our production department. 

Kind regards, 

on behalf of

Dr. Fares Alahdab 

Academic Editor

PLOS ONE